# Altered Regional Homogeneity and Amplitude of Low-Frequency Fluctuations Induced by Electroconvulsive Therapy for Adolescents with Depression and Suicidal Ideation

**DOI:** 10.3390/brainsci12091121

**Published:** 2022-08-24

**Authors:** Xiao Li, Xiaolu Chen, Yi Zhou, Linqi Dai, Long-Biao Cui, Renqiang Yu, Ming Ai, Qian Huang, Yu Tian, Mei Ming, Li Kuang

**Affiliations:** 1Department of Psychiatry, The First Affiliated Hospital of Chongqing Medical University, Chongqing 400016, China; 2The First Branch, The First Affiliated Hospital of Chongqing Medical University, Chongqing 400015, China; 3Department of Clinical Psychology, Fourth Military Medical University, Xi’an 710032, China; 4Department of Radiology, The First Affiliated Hospital of Chongqing Medical University, Chongqing 400016, China; 5Department of the First Clinical Medicine, Chongqing Medical University, Chongqing 400016, China

**Keywords:** MDD, adolescent, ALFF, ReHo, suicide ideation, electroconvulsive therapy

## Abstract

Resting-state functional magnetic resonance imaging (rs-fMRI) was used to investigate the effects of electroconvulsive therapy (ECT) causing brain function changes in adolescents who suffered from depression and suicidal ideation (SI). A total of 30 patients (MDDs) and 25 healthy controls (HCs) matched by gender, age, and education level were enrolled. The amplitude of low-frequency fluctuations (ALFF) and regional homogeneity (ReHo) were used to compare differences between HCs and MDDs at baseline, and differences in ALFF and ReHo pre/post ECT in MDDs. Pearson correlation analysis was used to evaluate the relationship between altered brain function and clinical symptoms. At baseline, MDDs showed decreased ALFF in the left inferior temporal gyrus and right amygdala, decreased ReHo in left inferior temporal gyrus, and increased ReHo in the right inferior frontal gyrus, opercular part and left middle occipital gyrus. After ECT, MDDs showed increased ALFF in the right middle occipital gyrus, decreased ALFF in left temporal pole, left inferior frontal gyrus, opercular part, and right frontal middle gyrus, increased ReHo in the right middle occipital gyrus, and left inferior temporal gyrus. Pearson correlation found HAMD scores at baseline were negatively correlated with ALFF in the left inferior temporal gyrus, and HAMD and BSSI scores after ECT were negatively correlated with ALFF in the right middle occipital gyrus. The abnormal activities of amygdala, inferior temporal gyrus and middle occipital gyrus might be related to depressive and suicidal symptoms in adolescents.

## 1. Introduction

Major depressive disorder (MDD) is a major source of impairment in adolescence and is considered to increase the risk of suicide, which is now the second leading cause of death in adolescents in America [1]. The lifetime prevalence of MDD among those aged between 13 and 18 is 11.0% in America [2], and every year, approximately 18–30% of adolescents have suicidal ideation (SI) [3,4]. SI is defined as “thoughts about death, dying, plans for suicide, or desire for death” [5,6], it is strongly predictive of death by suicide [7]. Zhang et al. [8] investigated adolescents in Beijing and found 12% of them had SI, with female/male = 13.3%/10.7%. Therefore, investigating SI in MDD adolescents is of necessity and significance.

Electroconvulsive therapy (ECT) is considered to be one of the best methods for MDD, and the remission rate could even reach 50–60% [9,10], especially for MDD with SI or suicidal attempt (SA) [11]. For its use on adolescents, it had been supported by the American Academy of Child and Adolescent Psychiatry (AACAP) since 2004 [12]. However, due to restrictions on research on adolescents and legal limitations [13,14], few large studies of ECT on adolescents are available, nonetheless, many studies still demonstrated that ECT could be considered amongst MDD adolescents accompanied by SI/SA [15,16].

Resting-state functional magnetic resonance imaging (rs-fMRI) is widely performed to investigate brain function in mental disorders. Amplitude of low frequency fluctuation (ALFF) and regional homogeneity (ReHo) are two fundamental rs-fMRI parameters describing local properties of resting-state brain function. ALFF can be used to test the magnitude of spontaneous blood-oxygen-level-dependent (BOLD) signal [17]. A systematic review found that ALFF of patients with MDD changed widely in the brain, such as orbital gyrus, precentral gyrus, etc. [18]. Lan et al. found that in MDD patients with SI, ALFF were higher in the right hippocampus, bilateral thalamus and caudate compared with patients without SI [19]. In addition, Liu et al. demonstrated that decreased depressive symptoms were negatively correlated with increased ALFF in the left hippocampus after eight ECT courses [20]. Regional homogeneity (ReHo) assumes that a given voxel is temporally similar to its neighbors, which is mainly used to detect spontaneous hemodynamic responses of rs-fMRI [21], and now has been widely used to detect the brain function in mental health disorders [22,23]. Kong et al. found decreased ReHo in the bilateral superior frontal gyrus after ECT in elder MDD patients [24].

However, studies on ECT for adolescents evaluated by rs-fMRI are limited. Thus, in our study, we have examined adolescents with MDD and SI who underwent ECT, and have observed the whole brain ALFF and ReHo pre/post ECT. Therefore, we have hypothesized that: (1) the adolescents with MDD and SI will show changed ReHo and ALFF compared with HCs; and (2) ECT would alter ALFF and ReHo in MDD adolescents with SI.

## 2. Materials and Methods

### 2.1. Subjects

All participants aged between 12 to 17 years were enrolled from April 2020 to March 2021. The diagnosis was confirmed by two senior psychiatrists using the Mini International Neuropsychiatric Interview for Children and Adolescents (MINI-KID). The inclusion criteria were as follows: (1) patients who fulfilled the MDD diagnostic criteria of the Diagnostic and Statistical Manual of Mental Disorders, Fourth Edition; (2) the Hamilton Depression Rating Scale (HAMD−17) ≥ 17; (3) first onset MDD or diagnosed before but without antidepressants in the last 8 weeks, no history of ECT treatment; and (4) patients with suicidal ideation (SI) in the last week. Patients were excluded if they had any history of psychiatric disorders such as bipolar disorder or schizophrenia, a previously diagnosed with organic brain diseases or other serious physical illnesses, history of substance abuse or dependence, or exhibited contraindications for MRI scanning.

HCs were volunteers who matched with the MDDs in gender, age and educational level. Controls reported neither lifetime psychiatric disorder nor a family history of psychosis in their first-degree relatives. Otherwise, the exclusion criteria remained the same as MDDs.

The study protocol has been approved by the Human Research and Ethics Committee of the First Affiliated Hospital of Chongqing Medical University (no. 2017-157). All adolescents and their caregivers had written informed consent after being told details of the study.

### 2.2. Clinical Assessment

The severity of symptoms was evaluated by Hamilton Depression Rating Scale [25] and the Beck Scale for Suicide Ideation [26] at baseline and after ECT. The Chinese version of HAMD/BSSI showed reliable and valid [27,28].

### 2.3. Electroconvulsive Therapy

The benefits and side effects of ECT were explained at the beginning of ECT sessions to every participant and their caregivers, hence obtaining their written informed consent. All ECT sessions were performed from 8:00 am to 12:00 am at the First Affiliated Hospital of Chongqing Medical University, using the Thymatron DGx (Somatics, LLC, Lake Bluff, IL, USA), with electrodes placed on the bilateral temporal lobes. We chose low 0.25 mode, 0.9A of electricity, the initial energy of ECT was generally selected as age × 50%, anesthesia and muscle relaxation were performed as follows: diprivan: 1.5–2 mg/kg, succinylcholine: 0.5–1 mg/kg. The ECT treatment course was taken once a day for the first 3–4 days, and then once every other day, with two days off in the weekends, every patient underwent ECT eight times, and the ECT sessions were completed after two weeks. All patients received medications during ECT sessions. (Details in Table 1)

### 2.4. Rs-fMRI Acquisition

A 3T GE Signa HDxt scanner (GE Healthcare, Chicago, IL, USA) equipped with an 8-channel head coil was used for rs-fMRI scanning. MDDs underwent scanning twice (pre/post ECT), and HCs once at baseline. Participants of HCs and MDDs underwent an MRI-scan between 6-8 pm, and the MDDs underwent an MRI-scan one day before ECT and the first day after ECT course ended, participants were directed to keep awake, stay relax and avoid thinking as much as possible. Head motion and machine noise were respectively mitigated by using foam pads and earplugs. All echo-planar imaging pulse sequence parameters were as followed: repetition time (TR) = 2000 ms, echo time (TE) = 40 ms, field of view (FOV) = 240 × 240 mm, matrix = 64 × 64, flip angle = 90°, slice number = 33, slice thickness/gap = 4.0/0 mm; scanner time = 8 min. Three-dimensional T1-weighted MR images used for rs-fMRI co-registration followed: TR = 24 ms; TE = 9 ms; FOV = 240 × 240 mm; matrix = 256 × 256; flip angle = 90°; slice thickness/gap = 1.0/0 mm.

### 2.5. Image Preprocessing

All data preprocessing was performed in MATLAB 2013b (MathWorks, Natick, MA, USA) using DPARSF (by YAN Chao-Gan, version 4.3, http://www.restfmri.net, accessed on 17 August 2022), which was based on SPM12. The main steps were as follows: (1) the initial 10 time points were discarded to ensure sufficient time for adaptation to the scanning process; (2) images were corrected for head movement and slice timing, if the head motion exceeding 2.5 mm along the x, y, or z axes or 2.5° in rotation, patients would be excluded; (3) the images were spatially normalized into standard Montreal Neurological Institute (MNI) space using the transformation derived from T1 segmentation and resampled at 3 × 3 × 3 mm^3^; (4) nuisance regression was performed using the 24 head motion parameters, white matter, and cerebrospinal fluid signals as covariates; (5) linear trends were removed; (6) the images were bandpass filtered to eliminate low-frequency drift and high-frequency noise.

### 2.6. Calculation of ReHo and ALFF

The ReHo was performed using the Kendall’s coefficient of concordance (KCC) of the time series of this voxel with its 26 nearest neighbors [29]. The raw ReHo of each voxel was then divided by the global mean ReHo value for each participant to reduce the global effects [30].

The rs-fMRI data was transformed to a frequency domain with a fast Fourier transformation from time series. The square root of the power spectrum was calculated, and ALFF obtained as the mean square root across 0.01–0.08 Hz. The ALFF value of each voxel was then divided by the global mean ALFF value for each subject to reduce the global effects. ALFF was computed as the mean power spectrum in a specific low-frequency band [31].

### 2.7. Statistical Analysis

Statistical analyses were conducted using SPSS (version26.0; IBM Corp., Armonk, NY, USA). Levene’s test and *t*-test were used to test the demographic characteristics and symptom scores. Two-sample *t*-test was conducted in SPM12 to compare ReHo and ALFF between MDDs pre-ECT and HCs, and paired *t*-tests were conducted between MDDs pre/post-ECT. The significance level was *p* value < 0.05 with Gaussian random field (GRF) correction. Pearson correlations were performed to examine the correlations between ReHo/ALFF in altered brain regions and clinical symptoms (HAMD/BSSI scores) pre/post ECT.

## 3. Results

### 3.1. Clinical Outcomes

The psychological measurements and demographic data are listed in Table 2. There were no significant differences between MDDs and HCs (*p* > 0.05). After ECT sessions, HAMD and BSSI scores significantly decreased (*p* < 0.05) (Table 3).

### 3.2. Neuroimaging Comparisons between MDDs at Baseline and HCs

Compared with HCs, decreased ALFF was found in the right amygdala (amygdala _R) and the left inferior temporal gyrus (Temporal_Inf_L) of MDDs, increased ReHo was found in right inferior frontal gyrus, opercular part (Frontal_Inf_Oper_R) and left middle occipital gyrus (Occipital_Mid_L) of the MDDs, and decreased ReHo was found in left inferior temporal gyrus (Temporal_Inf_L) of the MDDs (Table 4, Figure 1 and Figure 2).

### 3.3. Neuroimaging Comparisons in MDDs Pre/Post-ECT

Compared with pre-ECT, the ALFF in the right middle occipital gyrus (Occipital_Mid_R) increased, ALFF in the left temporal pole: superior temporal gyrus (Temporal_Pole_Sup_L), the left inferior frontal gyrus, opercular part (Frontal_Inf_Oper_L), and the right frontal middle gyrus (Frontal_Mid_R) decreased post-ECT, ReHo in the right middle occipital gyrus (Occipital_Mid_R), and the left inferior temporal gyrus (Temporal_Inf_L) increased post ECT (Table 5, Figure 3 and Figure 4).

### 3.4. Pearson Correlation Analysis

Pearson correlation analysis found that BSSI scores at baseline was negatively correlated with the ALFF in the Temporal_Inf_L (Figure 5A,B), and the HAMD and BSSI scores after ECT was negatively correlated with the ALFF in the Occipital_Mid_R (Figure 6A,B and Figure 7A,B).

## 4. Discussion

This study found that after 2 weeks of ECT, the HAMD/BSSI scores of adolescents with MDD and SI significantly reduced, indicating that ECT could quickly alleviate the depression and suicide symptoms of adolescent patients. We used ReHo values and found at baseline, MDDs displayed decreased ReHo in the Temporal_Inf_L, and increased ReHo in the Frontal_Inf_Oper_R and Occipital_Mid_L, after ECT, increased ReHo in the Occipital_Mid_R and Temporal_Inf_L were found. By means of ALFF values, we found a decreased ALFF in the Temporal_Inf_L and Amygdala_R. After ECT, MDDs showed increased ALFF in the Occipital_Mid_R, decreased ALFF in the Temporal_Pole_Sup_L, the Frontal_Inf_Oper_L, and the Frontal_Mid_R, which showed the changes of ReHo and ALFF values mainly distributed in the frontal lobe, occipital lobe and temporal lobe.

In our study, we found that brain function changed in the prefrontal cortex pre/post ECT in MDDs. The prefrontal cortex is an important brain area involving multiple functions such as emotion, cognition, memory, and reward. Memory loss or inattention of MDD patients may be related to the disfunction of the prefrontal cortex. When the expected reward is not received, the patient may be more disappointed than healthy people. At the same time, the prefrontal cortex is also correlated with the brain area of self-perception, if the patient does not receive reward feedback, it may lead to personal loss and low self-esteem and even suicidal ideation, whereby, according to previous studies findings, there was a significantly reduced prefrontal cortex volume in MDD patients with high suicide risk [32,33]. Willeumier et al. found that the perfusion of the bilateral upper and medial prefrontal cortex decreased in MDD patients, and it was believed that the impairment function of the prefrontal cortex was closely related to the occurrence of suicide [34]. Jollant et al. reported a decreased activity of the prefrontal lobe in depression patients with suicide attempts (SA) when faced with angry faces [35,36]. Pan et al. also found that, compared with HCs and MDD male adults without SA, MDD male adults with SA showed decreased activity of the prefrontal lobe when facing angry faces [37]. These studies showed suicide attempters presenting higher sensitivity to negative emotions, and reduced attention to positive emotions, and these suicidal symptoms were correlated with prefrontal cortex function. One study focused on the structural changes of the ventrolateral prefrontal cortex (VLPFC) in patients with MDD and SI, and found that the grey matter volume (GMV) of the right VLPFC in patient group significantly reduced compared with HCs [38], suggesting that the structure of VLPFC was associated with SI. Another study evaluated MDD adolescents with SI by functional near-infrared spectroscopy (fNIRS) and found the decreased activity of the left VLPFC in the SI group compared with HCs, and the altered oxyhemoglobin were associated with the severity of SI [39]. Our study found that altered brain function of VLPFC in the patients pre/post ECT, suggesting that abnormal function in this area might be leading to the depression and suicidal symptoms, and ECT may regulate this area to achieve the occurrence of depression and suicide.

Abnormalities in the middle occipital gyrus (Occipital_Mid) have also been studied a lot in MDD patients, and are considered to play an important role in the perception of emotional facial expression stimuli. Therefore, the abnormal structure and function of the Occipital_Mid may lead to depression in patients. For example, Liu et al. found that the bilateral occipital cortex reduced in MDD adults [40]. A meta-analysis focused on the functional connectivity (FC) study of patients with suicidal symptoms showed that individuals with suicidal symptoms had altered brain function in bilateral Occipital_Mid [41]. Another study also found that, compared with HCs, depressed patients with suicidal symptoms showed reduced ALFF in the Occipital_Mid_L [42]. When focusing on female depression, it was found that the ALFF value in the Occipital_Mid_L decreased in comparison with that of HCs [43], suggesting that abnormal ALFF in the Occipital_Mid was closely correlated with depressive symptoms. In addition to the change in ALFF, we found compared with HCs, ReHo of the Occipital_Mid_L in the MDDs increased. Zhang et al. [44] found decreased ReHo in Occipital_Mid_L in MDDs, which was contrary to that of our results, where their study’s focus on female MDD patients and a different sample might be the cause. However, it could still indicate the special status of ReHo in the Occipital_Mid in depression patients. We found both ALFF and ReHo were increased in the Occipital_Mid_R after ECT. A study showed decreased ReHo values in the Occipital_Mid_R in MDD patients [45]. In our study, ReHo of the Occipital_Mid_R was increased after ECT, and both HAMD and BSSI scores after ECT significantly correlated with ALFF in Occipital_Mid_R. These results showed ALFF changed in Occipital_Mid_R which might be related to clinical outcome.

In our study, both ALFF and ReHo in the Temporal_Inf_L of MDDs at baseline decreased compared with HCs, while ReHo changed significantly after ECT, suggesting that changes of brain function in the Temporal_Inf_L played an important role in adolescent depression. Previous studies found that the temporal lobe could participate in the recall of personal experiences, and predict individual behavioral abilities based on personal beliefs and emotions [46,47]. Therefore, abnormal brain activity in the temporal lobe may lead to emotional dysregulation and increase suicide risk in depression. Previous studies also found abnormal brain function in the superior temporal gyrus of MDD adolescents with suicidal behavior [42,48]. In our study, brain function changes in the Temporal_Inf_L were found, changes in the Temporal_Inf_L were common in early-onset MDD patients, where this result was also found by Ramezani et al. in the altered Temporal_Inf_L in the early-onset of depression [49]. Another finding relating to the Temporal_Inf_L was the relationship with childhood trauma. For example, Du et al. found that, compared with HCs, there was decreased fALFF in Temporal_Inf_L in MDD patients with childhood trauma [50], therefore, we consider that the Temporal_Inf_L has guiding significance in adolescent subtype depression. ECT may improve depressive symptoms by activating local brain function in the Temporal_Inf_L.

Our study also found decreased ALFF in the right amygdala of MDDs compared with HCs. The amygdala is a component of the limbic system of the brain and is involved in emotion recognition and regulation. Previous studies found the amygdala could enhance pleasant or unpleasant experiences through emotional regulation [51]. Rs-fMRI results demonstrated that the amygdala was both structurally and functionally connected to brain regions related to emotion regulation in MDD patients, such as the hippocampus, anterior cingulate gyrus, and orbitofrontal lobe [52]. In adolescent depression, studies found decreased resting-state functional connectivity (rs-FC) between the amygdala and hippocampus, which was associated with loss of pleasure, low mood, irritability, and burnout [53]. In this study, we found decreased ALFF in the right amygdala in the MDDs compared with HCs, suggesting MDD adolescents with SI still had abnormal amygdala function.

This study has some limitations. First, the sample size of the study was relatively small, and further studies with larger samples are needed to verity our findings. Second, there was lack of MDD without SI group to compare the local brain function with the MDD with SI group, so this study could not distinguish the effect of ECT for MDD and SI. Third, all the patients continued their medications throughout the ECT series since this was ethically necessary. Thus, the observed effects are not only caused by ECT. Fourth, we only perform one measurement in HCs, so the changes in brain function in MDDs before and after treatment were not specifically attributable to ECT without correction in HCs, in a future study, we could also perform a second measurement kept consistent with MDDs in HCs.

## 5. Conclusions

This study focused on adolescents with MDD and SI, we found ReHo and ALFF changes between the brains of MDDs and HCs, and brain functions of Occipital_Mid_R, Temporal_Pole_Sup_L, Frontal_Inf_L, Frontal_Mid_R and Temporal_Inf_L changed after ECT. ECT may have regulatory effects on brain regions to compensate for original defects.

## Figures and Tables

**Figure 1 brainsci-12-01121-f001:**
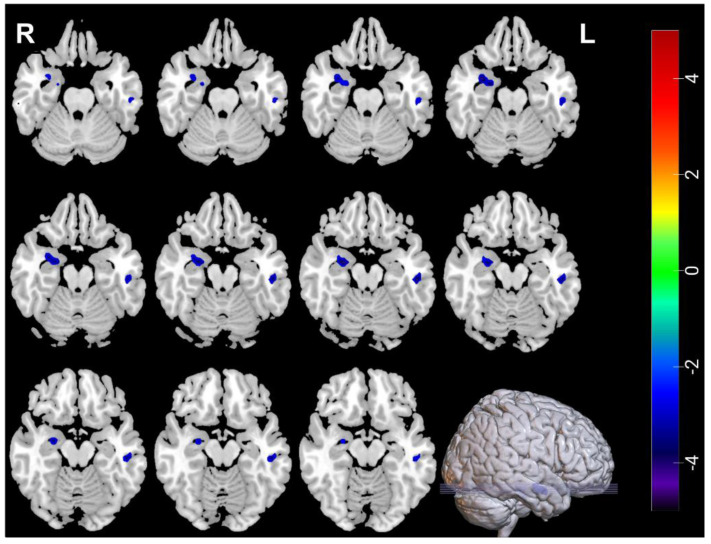
The blue regions mean decreased ALFF in Amygdala_R and Temporal_Inf_L between MDDs at baseline and HCs. ALFF voxel *p* < 0.005; GRF correction *p* > 0.05; cluster size > 24.

**Figure 2 brainsci-12-01121-f002:**
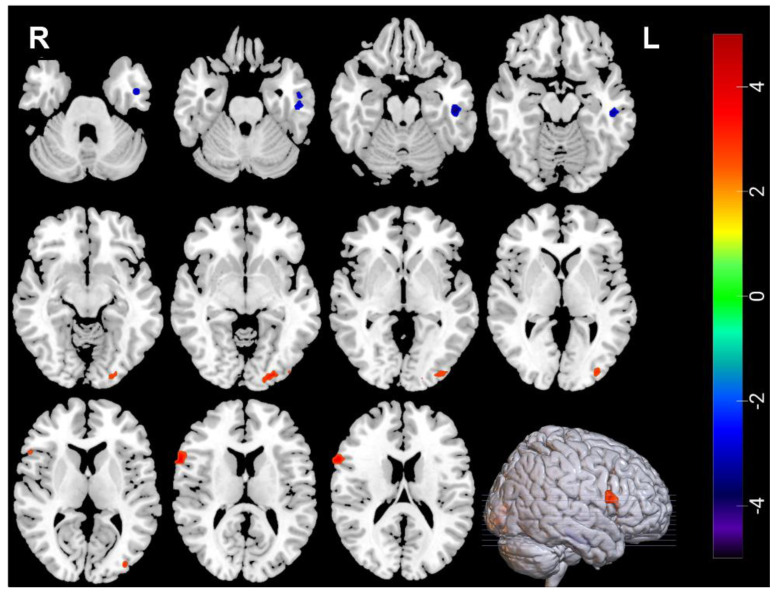
The blue regions represent decreased ReHo in MDDs at baseline compared with HCs in Temporal_Inf_L, the red regions represent increased ReHo in MDDs at baseline compared with HCs in Frontal_Inf_Oper_R and Occipital_Mid_L. ReHo voxel *p* < 0.005; GRF correction *p* < 0.05; cluster size > 34.

**Figure 3 brainsci-12-01121-f003:**
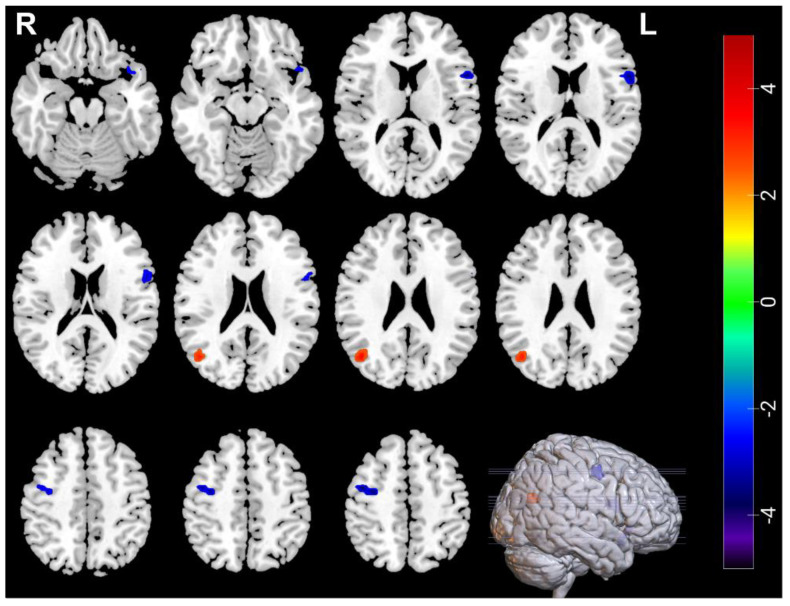
The blue regions represent decreased ALFF in Temporal_Pole_Sup_L, Frontal_Inf_Oper_L and Frontal_Mid_R after ECT, the red regions represent increased ALFF in Occipital_Mid_R in MDDs after ECT. ALFF voxel *p* < 0.005; GRF correction *p* < 0.05; cluster size > 24.

**Figure 4 brainsci-12-01121-f004:**
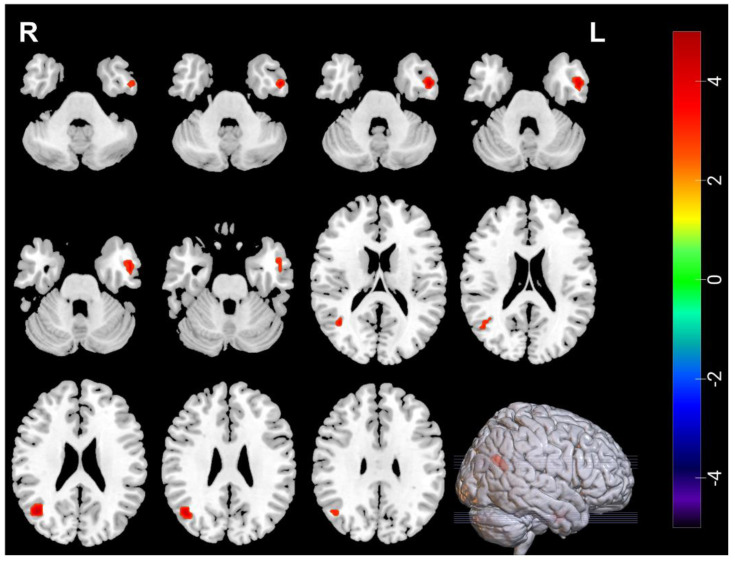
The red regions represent increased ReHo in Temporal_Inf_L and Occipital_Mid_R in MDDs after ECT. ReHo voxel *p <* 0.005; GRF correction *p <* 0.05; cluster size > 34.

**Figure 5 brainsci-12-01121-f005:**
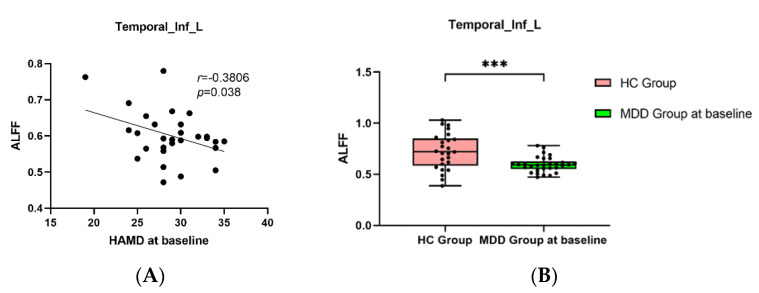
(**A**): The negative correlations between HAMD scores at baseline and ALFF in the Temporal_Inf_L (*p* = 0.038). (**B**): MDDs showed decreased ALFF in Temporal_Inf_L compared with HCs (*** *p* < 0.001).

**Figure 6 brainsci-12-01121-f006:**
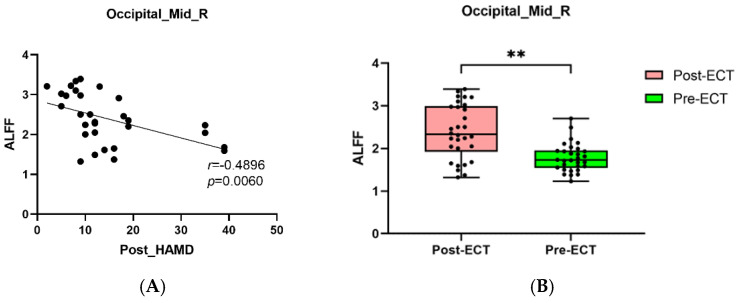
(**A**): The negative correlations between HAMD scores after ECT and ALFF in the Occipital_Mid_R (*p* = 0.0060). (**B**): MDDs showed increased ALFF in Occipital_Mid_R after ECT (** *p* < 0.01).

**Figure 7 brainsci-12-01121-f007:**
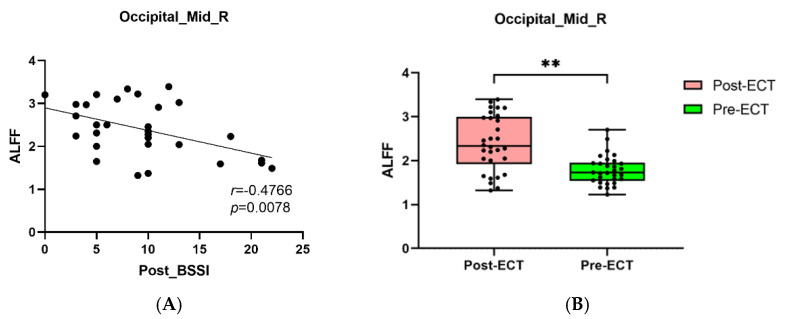
(**A**): The negative correlations between BSSI scores after ECT and ALFF in the Occipital_Mid_R (*p* = 0.0078). (**B**): MDDs showed increased ALFF in Occipital_Mid_R after ECT (** *p* < 0.01).

**Table 1 brainsci-12-01121-t001:** Medications in MDDs.

Types of Medications	
Antidepressants (n = 30)	Sertraline (n = 17)
	Fluoxetine (n = 11)
	Venlafaxine (n = 1)
	Escitalopram (n = 1)
Antipsychotics (n = 26)	Quetiapine (n = 13)
	Olanzapine (n = 8)
	Aripiprazole (n = 4)
	Risperidone (n = 1)
Anti-anxiety drugs (n = 5)	Tandospirone (n = 5)
Others (n = 3)	Propranolol (n = 2)
	TrihexyPhenidyl (n = 1)

**Table 2 brainsci-12-01121-t002:** Demographics and clinical characteristics between MDDs and HCs.

Characteristic	HC (n = 25)	MDD (n = 30)	*t*-Value	*p*
Age, mean (SD), y	15.48 (1.87)	14.60 (1.45)	1.962	0.197 *
Sex (male/female)	6/19	8/22	0.222	0.657 ^#^
Education years, mean (SD), y	9.68 (2.21)	8.50 (1.70)	0.143	0.143 *
HAMD, mean (SD)	1.60 (2.06)	29.03 (6.02)	/	<0.001 *
BSSI, mean (SD)	0	22.10 (5.73)	/	<0.001 *

Note: MDD, major depressive disorder; HC, healthy control; HAMD, Hamilton depression scale; BSSI, Beck scale for suicide ideation; SD, Standard deviation. * Two sample t-test; ^#^ Chi-square test.

**Table 3 brainsci-12-01121-t003:** Comparison of HAMD/BSSI scores pre/post-ECT.

Characteristic	Pre-ECT	Post-ECT	*t*-Value	*p*
HAMD, mean (SD)	29.03(6.02)	13.77(8.89)	9.762	<0.001 *
BSSI, mean (SD)	22.10(5.73)	8.10(6.94)	10.734	<0.001 *

Note: ECT: Electroconvulsive Therapy; HAMD, Hamilton depression scale; BSSI, Beck scale for suicide ideation; SD, Standard deviation. * Paired *t*-test.

**Table 4 brainsci-12-01121-t004:** Significant differences in ALFF and ReHo between MDDs and HCs.

Measures	Brain Regions	Voxel Size	Peak *t* Value	MNI Coordinates		
**Decreased**						
ALFF	Amygdala_R	31	−4.637	33	0	−21
ALFF	Temporal_Inf_L	30	−3.6166	−51	−24	−18
ReHo	Temporal_Inf_L	45	−4.3107	−48	−21	−18
**Increased**						
ReHo	Frontal_Inf_Oper_R	40	4.0503	63	15	15
ReHo	Occipital_Mid_L	41	3.7464	−36	−84	6

Note: ALFF: amplitude of low frequency fluctuations; ReHo: regional homogeneity; MNI: Montreal Neurological Institute.

**Table 5 brainsci-12-01121-t005:** Significant differences in ALFF and ReHo of MDDs pre/post ECT.

Measures	Brain Regions	Voxel Size	Peak *t* Value	MNI Coordinates		
**Decreased**						
ALFF	Temporal_Pole_Sup_L	30	−4.0615	−51	18	−15
ALFF	Frontal_Inf_Oper_L	39	−3.9169	−54	12	12
ALFF	Frontal_Mid_R	34	−4.2384	42	−3	51
**Increased**						
ALFF	Occipital_Mid_R	30	3.7765	42	−66	24
ReHo	Occipital_Mid_R	46	4.5719	42	−63	24
ReHo	Temporal_Inf_L	38	4.761	−51	−3	−33

Note: ALFF: amplitude of low frequency fluctuations; ReHo: regional homogeneity; MNI: Montreal Neurological Institute.

## Data Availability

The original contributions presented in the study are included in the article, further inquiries can be directed to the corresponding author/s.

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
