# Peer review of "Altered Regional Homogeneity and Amplitude of Low-Frequency Fluctuations Induced by Electroconvulsive Therapy for Adolescents with Depression and Suicidal Ideation"

_brainsci, 2022, doi:10.3390/brainsci12091121_

Round 1

Reviewer 1 Report

The authors present an important topic regarding young patients (12-17 yo) with depressive symptoms, who were treated with ECT. The authors try to associate ECT-outcomes on depressive symptomatology and suicidal ideation with fMRI-calcilations (ALFF and ReHo). Although this topic needs scientific attention, I think the authors did not succeed in proper answers to their research questions.

First, examining such young patients, especially when treated with ECT (which should be very rare in these cases), require very special ethical attention. The authors only state that written informed consent from all 'adolescents' (sometimes 12 yo are still young children) and their caregivers was given, which is rather limited. Also, diagnosing severe depressieve disorders in such young patients is challenging. Also, suicidal ideation may be different than in adult patients, especially when they use SSRIs (n=30, all included MDD patients!), which may have increased suicidality. How was the diagnostic process described? Which tools were used? Were the SSRIs discontinued and evaluated whether the suicidality decreased?

Secondly, may major methodological concern is that the authors did not used follow-up scans in the HCs. They compare MDDs and HCs at baseline, followed by firm conclusions about changes in ALFF and ReHo after ECT. But, if these results were not corrected for changes in HCs in time, their results are only interesting descriptions but certainly no proof of influence of ECT on those fMRI-parameters. No word of this very large limitations in their (rather 'modestly' written) limitation section.

Some other comments:

- the authors do not describe what the total ECT-sessions were during the course of the patients. What happened if the patient did not recover after the course, etc.;

- which ECT-parameters were used (e.g., pulsewidth, thymatron uses 0.9 A);

- some English need correction.

Author Response

Response to Reviewer 1 Comments

The authors present an important topic regarding young patients (12-17 years) with depressive symptoms, who were treated with ECT. The authors try to associate ECT-outcomes on depressive symptomatology and suicidal ideation with fMRI-calcilations (ALFF and ReHo). Although this topic needs scientific attention, I think the authors did not succeed in proper answers to their research questions.

First, examining such young patients, especially when treated with ECT (which should be very rare in these cases), require very special ethical attention. The authors only state that written informed consent from all 'adolescents' (sometimes 12 years are still young children) and their caregivers was given, which is rather limited. Also, diagnosing severe depressive disorders in such young patients is challenging. Also, suicidal ideation may be different than in adult patients, especially when they use SSRIs (n=30, all included MDD patients!), which may have increased suicidality. How was the diagnostic process described? Which tools were used? Were the SSRIs discontinued and evaluated whether the suicidality decreased?

-Response: Thanks for the suggestion, adolescents who took ECT were always carefully evaluated, from supports by the American Academy of Child and Adolescent Psychiatry (AACAP) since 2004, adolescent MDD with suicidal ideation could be consider to take ECT. These adolescents enrolled in our study were found with severe suicidal ideation in recent one week, and before we started our study, we got approvement from the Human Research and Ethics Committee of the First Affiliated Hospital of Chongqing Medical University (no. 2017-157), and written informed consent from participants and their caregivers.

The diagnosis was based on DSM-IV and M.I.N.I-KID, evaluated by two experienced psychiatrists, the depressive symptom was evaluated by Hamilton Depression Rating Scale and the suicidal ideation was evaluated by Beck Scale for Suicide Ideation, we have added the information in the “subjects” section, it was in page 2. Line 33-34. In our study, we tried to observe ECT induced brain function change, so we didn.t evaluate the SSRIs when the suicidality decreased, this is our limitation.

Secondly, may major methodological concern is that the authors did not used follow-up scans in the HCs. They compare MDDs and HCs at baseline, followed by firm conclusions about changes in ALFF and ReHo after ECT. But, if these results were not corrected for changes in HCs in time, their results are only interesting descriptions but certainly no proof of influence of ECT on those fMRI-parameters. No word of this very large limitations in their (rather 'modestly' written) limitation section.

- Response: Thanks for the suggestion, in our research, we didn’t correct for changed in HCs after 2 weeks, so the changed in MDDs might be caused by ECT and some other factors, this is our limitation, we have corrected our conclusion, and we have added them in the “limitation” section, it was in page 10, line 38-39.

Some other comments:

- the authors do not describe what the total ECT-sessions were during the course of the patients. What happened if the patient did not recover after the course, etc.;

- Response: Thanks for the suggestion, we have added the ECT-sessions in the “electroconvulsive therapy” session, it is in page 3, line 14, in our research, we tried to find out the ECT induced ReHo and ALFF changes in ECT sessions, and we didn’t discuss information about these patients who did not recover after the course. This is our limitation. Thanks for your suggestion.

- which ECT-parameters were used (e.g., pulsewidth, thymatron uses 0.9 A);

- Response: Thanks for the suggestion, we have added the information of ECT-parameters in the “electroconvulsive therapy” section. It is in page 3, line 9-11.

- some English need correction.

- Response: Thanks for the suggestion, we have asked native speaker to check the language and also grammar mistake.

Reviewer 2 Report

The manuscript provides an interesting approach to evaluate the effect of electroconvulsive therapy on regional homogeneity and the amplitude of low-frequency fluctuations in resting state functional magnetic resonance imaging in patients with major depressive disorder.

The overall manuscript need language and grammar editing (upper/lower case, fALFF vs ALFF, punctuation) as well as formal editing to fit the Journals formal requirements (Table/Figure Legends and so on).

Abbreviations should be introduced at first use in the abstract and the manuscript text. I would also prefer to not use abbreviations in the title.

The introduction provides a sufficient overview on the topic. However, the authors mix up fMRI and rs-fMRI in the text. Please specify. Has ALFF and ReHo been investigated to be reliable measures over time (HC studies)?

Regarding the acquisition of rs-fMRI data, was data acquires ascending/descending or interleaved? Was rs-fMRI performed prior or after T1-weighted image acquisition? Was there any preparation for the patient to get used to the scanner environment or so to reduce external bias?

Image processing encompassed the standard pipeline. However it remains unclear how the T1-weighted images are included. In a standard workflow one would co-register the EPI images to the T1-weighted structural data set, and normalize the T1-weighted image set to the MNI space and transform the EPI data accordingly. Please specify.

For statistical analysis, were requirements for the applications of the named statistical tests performed (e.g. for t-test, homogeneity of variances tested using the Levene’s test?)

Regarding table 2. The text states that HAMD and BSSI were assessed in patients. The Table shows assessment of HAMD in HC and MDD, BSSI only in MDD. What was tested here, as a p-value is given for both?

Please revise Figure 7 (format. A,B).

Changes in ReHo and ALFF in relation to ECT are discussed. However, as also HC are investigated, are ReHo and ALFF anyhow reliable measures in rs-fMRI or are differences also seen in the HC group without intervention within the same follow-up interval? Please add more information here. How was MRI follow up scheduled (time interval, and so on). Was it always on the same time of the day, with asking for a similar behavior as this can tremendously impact the rs-fMRI measures?

Author Response

Response to Reviewer 2 Comments

The manuscript provides an interesting approach to evaluate the effect of electroconvulsive therapy on regional homogeneity and the amplitude of low-frequency fluctuations in resting state functional magnetic resonance imaging in patients with major depressive disorder.

The overall manuscript need language and grammar editing (upper/lower case, fALFF vs ALFF, punctuation) as well as formal editing to fit the Journals formal requirements (Table/Figure Legends and so on).

-Response: Thanks for the suggestion, we have asked help from native speaker to check language and grammar mistake, and corrected the formal to fit the Journals formal requirements

Abbreviations should be introduced at first use in the abstract and the manuscript text. I would also prefer to not use abbreviations in the title.

-Response: Thanks for the suggestion, we have revised the title and checked the whole paper about the abbreviations.

The introduction provides a sufficient overview on the topic. However, the authors mix up fMRI and rs-fMRI in the text. Please specify. Has ALFF and ReHo been investigated to be reliable measures over time (HC studies)?

- Response: Thanks for your suggestion, we have checked the mix up of fMRI and rs-fMRI in the whole paper, and in our study, we didn’t re-calculate the ALFF and ReHo of HCs, this is our limitation, we also mentioned it in the “limitation” section.

Regarding the acquisition of rs-fMRI data, was data acquires ascending/descending or interleaved? Was rs-fMRI performed prior or after T1-weighted image acquisition? Was there any preparation for the patient to get used to the scanner environment or so to reduce external bias?

- Response: Thanks for your question, the data acquires by interleaved, the rs-fMRI performed after T1-weighted image acquisition. The preparation we do as follows: 1) we scaned T1 first and then rs-fMRI to make sure the patient to get used to the scanner environment; 2) before MRI scan, we would tell patients the whole process, foam pads and earplugs were used to keep them feel relax and comfortable, make sure the participants get used to environment.

Image processing encompassed the standard pipeline. However, it remains unclear how the T1-weighted images are included. In a standard workflow one would co-register the EPI images to the T1-weighted structural data set, and normalize the T1-weighted image set to the MNI space and transform the EPI data accordingly. Please specify.

- Response: Thanks for the suggestion, we have revised the part, which is in the “image preprocessing” section. It is in page 4, line 7-9.

For statistical analysis, were requirements for the applications of the named statistical tests performed (e.g. for t-test, homogeneity of variances tested using the Levene’s test?)

- Response: Thanks for the suggestion, we do Levene’s test before the statistical analysis.

Regarding table 2. The text states that HAMD and BSSI were assessed in patients. The Table shows assessment of HAMD in HC and MDD, BSSI only in MDD. What was tested here, as a p-value is given for both?

- Response: Thanks for the suggestion, the BSSI score of each health participant is 0, we miswrite it to “/”, and here the statistical analysis is also two sample t-test, and p-value is given for both.

Please revise Figure 7 (format. A,B).

- Response: Thanks for the suggestion, we have revised Figure 7.

Changes in ReHo and ALFF in relation to ECT are discussed. However, as also HC are investigated, are ReHo and ALFF anyhow reliable measures in rs-fMRI or are differences also seen in the HC group without intervention within the same follow-up interval? Please add more information here. How was MRI follow up scheduled (time interval, and so on). Was it always on the same time of the day, with asking for a similar behavior as this can tremendously impact the rs-fMRI measures?

- Response: Thanks for the suggestion, in our study, we didn’t correct for changed in HCs after 2 weeks, this is our limitation, we have added them in the “limitation” section, and we added the information about MRI scheduled and the quality control during MRI scan in the “Rs-fMRI acquisition” section, it is in page 3, line 36-39.

Round 2

Reviewer 1 Report

Although the authors responded to my former comments, the adjustments they made in their current manuscript do not meet my fundamental concerns.

These concerns regard the clinical treatment of 30 very young patients, as the ECT-courses seem not to be applied in accordance with international guidelines. E.g., the authors describe daily treatments, only a maximum of 8 treatments, and inappropriate dosing at 70% of age with BL electrode placement. Also, including 30 of such young patients within one year, in one institute, seems a very high number to me, because adolescents would be treated with ECT very very infrequently. Moreover, my concerns regard the methodology, which is still not correct because of the lack of follow-up MRIs in the HCs. The authors fail to describe all methodological limitations in their very short (too short in my opinion) paragraph about these issues.

Author Response

Although the authors responded to my former comments, the adjustments they made in their current manuscript do not meet my fundamental concerns.

These concerns regard the clinical treatment of 30 very young patients, as the ECT-courses seem not to be applied in accordance with international guidelines. E.g., the authors describe daily treatments, only a maximum of 8 treatments, and inappropriate dosing at 70% of age with BL electrode placement.

-Response: Thanks for the question, we used BL electrode placement based on the severity of symptoms, and some previous studies showed unilateral ECT may be less effective and the memory impairment similar to that associated with bilateral ECT (1), and here we used low0.25 mode (relatively low cognitive impairment) which was thought suitable for adolescents. For the 8 treatments of ECT, we just wanted to investigate the controlled ECT sessions used for adolescents, and rescanned MRI when 8 treatments were over, after that, we would continue treatment depends on the clinical response, the energy was 50%, we apologize for our wrong writing.

  1. Sackeim HA, Prudic J, Devanand D et al. (1993), Effects of stimulus intensity and electrode placement on the efficacy and cognitive effects of electroconvulsive therapy. N Engl J Med 328:839–846.

Also, including 30 of such young patients within one year, in one institute, seems a very high number to me, because adolescents would be treated with ECT very very infrequently.

-Response: Thanks for the question, adolescents who take ECT should always be assessed very carefully, the use of ECT for Chinese adolescents is relatively common, yu-tao xiang et al (1) investigated 954 adolescents aged between 13 and 17 years over a period of 8 years (2007–2013) in a university-affiliated teaching psychiatric hospital, and found 42.6% of them took ECT. Previous studies showed increased knowledge and expertise and the reduction of complications associated with ECT may have contributed to the frequent use of ECT in China (2). In recent 10 years, the prevalence of MDD in China increase rapidly, a recent study published in 2022 showed the number of MDD adolescents under 18 years is more than 15,000,000 (3), at the same time, as a tertiary referral center in southwest of China, we received the most complex patients who are more likely to meet the clinical indication for ECT. The department of psychiatry in the hospital has 100 beds, receives approximate 420 outpatients visit daily, so a considerable proportion of adolescents could meet the criteria to take ECT.

  1. Zhang QE, Wang ZM, Sha S et al. Common Use of Electroconvulsive Therapy for Chinese Adolescent Psychiatric Patients. J ECT. 2016; 32(4):251-255.
  2. Leung CM, Xiang YT, He JL, et al. Modified and unmodified electroconvulsive therapy: a comparison of attitudes between psychiatrists in Beijing and Hong Kong. J ECT. 2009; 25:80–84.
  3. <blue book for national depression in 2022>Peopledailyhealth.https://m.peopledailyhealth.com/articleDetailShare?articleId=026345f72bbf4e9eaf312ff0e237ed51

Moreover, my concerns regard the methodology, which is still not correct because of the lack of follow-up MRIs in the HCs. The authors fail to describe all methodological limitations in their very short (too short in my opinion) paragraph about these issues.

-Response: We sincerely thank for the suggestion about our methodology, we pay not enough attention on using HC data and also test-retest data because of there are still many studies only scan HC once (1-3), but we also find more and more studies focused on ECT using HC data and test-retest data in recent years (4,5). Here we have added more limitation based on our knowledge.

  1. Wei Q, Bai T, Brown EC. et al. Thalamocortical connectivity in electroconvulsive therapy for major depressive disorder. J Affect Disord. 2020; 264:163-171.
  2. Qiu H, Li X, Luo Q. et al. Alterations in patients with major depressive disorder before and after electroconvulsive therapy measured by fractional amplitude of low-frequency fluctuations (fALFF). J Affect Disord. 2019; 244:92-99.
  3. Li XK, Qiu HT, Hu J.et al. Changes in the amplitude of low-frequency fluctuations in specific frequency bands in major depressive disorder after electroconvulsive therapy. World J Psychiatry. 2022 May 19;12(5):708-721.
  4. Enneking V, Dzvonyar F, Dück K. et al. Brain functional effects of electroconvulsive therapy during emotional processing in major depressive disorder. Brain Stimul. 2020 Jul-Aug;13(4):1051-1058.
  5. Tsolaki E, Narr KL, Espinoza R. et al. Subcallosal Cingulate Structural Connectivity Differs in Responders and Nonresponders to Electroconvulsive Therapy. Biol Psychiatry Cogn Neurosci Neuroimaging. 2021 Jan;6(1):10-19.

Reviewer 2 Report

Revision did improve the overall quality of the manuscript. However, there are still some aspects left:

Even though the authors state in the Response, that e.g. Levene's test was done, those results are missing in the manuscript underpinning the applicability of the t-tests that are performed.

The aspect of missing test-retest data should be further discussed, as this is a major drawback of the study.  Are there other studies dealing with similar topics using HC data and also test-retest data? Else the interpretation is pretty limited.

Author Response

Revision did improve the overall quality of the manuscript. However, there are still some aspects left:

Even though the authors state in the Response, that e.g. Levene's test was done, those results are missing in the manuscript underpinning the applicability of the t-tests that are performed.

-Response: Thanks for the suggestion, here we have added the T value in Table 1 and Table 2, and we also added the “Levene's test” in the “statistical analysis” section.

The aspect of missing test-retest data should be further discussed, as this is a major drawback of the study.  Are there other studies dealing with similar topics using HC data and also test-retest data? Else the interpretation is pretty limited.

-Response: Thanks for your suggestion, the drawback of the study we made based on similar studies like us (HC scanned once and patients scanned twice before and after ECT) were still common (1-3), but after we reviewed recent literatures, we found more and more studies focused on ECT using HC data and test-retest data (4,5), so we added more limitation based on our knowledge.

  1. Wei Q, Bai T, Brown EC. et al. Thalamocortical connectivity in electroconvulsive therapy for major depressive disorder. J Affect Disord. 2020; 264:163-171.
  2. Qiu H, Li X, Luo Q. et al. Alterations in patients with major depressive disorder before and after electroconvulsive therapy measured by fractional amplitude of low-frequency fluctuations (fALFF). J Affect Disord. 2019; 244:92-99.
  3. Li XK, Qiu HT, Hu J.et al. Changes in the amplitude of low-frequency fluctuations in specific frequency bands in major depressive disorder after electroconvulsive therapy. World J Psychiatry. 2022 May 19;12(5):708-721.
  4. Enneking V, Dzvonyar F, Dück K. et al. Brain functional effects of electroconvulsive therapy during emotional processing in major depressive disorder. Brain Stimul. 2020 Jul-Aug;13(4):1051-1058.
  5. Tsolaki E, Narr KL, Espinoza R. et al. Subcallosal Cingulate Structural Connectivity Differs in Responders and Nonresponders to Electroconvulsive Therapy. Biol Psychiatry Cogn Neurosci Neuroimaging. 2021 Jan;6(1):10-19.
